**Data Availability Statement:** The data may be found in the website: www.covid19india.org.

**Funding:** The author(s) received no specific funding for this work.

# District level correlates of COVID-19 pandemic in India during March-October 2020

Vandana Tamrakar[1], Ankita Srivastava[1], Nandita Saikia[1]*, Mukesh C. Parmar[1], Sudheer Kumar Shukla[1], Shewli Shabnam[2], Bandita Boro[1], Apala Saha[3], Benjamin Debbarma[1]

1 Centre for the Study of Regional Development, Jawaharlal Nehru University, New Delhi, India, 2 Bidhannagar College, Kolkata, India, 3 Banaras Hindu University, Varanasi, India

* nanditasts@gmail.com

## Abstract

### Background

COVID-19 is affecting the entire population of India. Understanding district level correlates of the COVID-19's infection ratio (IR) is essential for formulating policies and interventions.

### Objective

The present study aims to investigate the district level variation in COVID-19 during March-October 2020. The present study also examines the association between India's socioeconomic and demographic characteristics and the COVID-19 infection ratio at the district level.

### Data and methods

We used publicly available crowdsourced district-level data on COVID-19 from March 14, 2020, to October 31, 2020. We identified hotspot and cold spot districts for COVID-19 cases and infection ratio. We have also carried out two sets of regression analysis to highlight the district level demographic, socioeconomic, household infrastructure facilities, and health-related correlates of the COVID-19 infection ratio.

### Results

The results showed on October 31, 2020, the IR in India was 42.85 per hundred thousand population, with the highest in Kerala (259.63) and the lowest in Bihar (6.58). About 80 percent infected cases and 61 percent deaths were observed in nine states (Delhi, Gujarat, West Bengal, Uttar Pradesh, Andhra Pradesh, Maharashtra, Karnataka, Tamil Nadu, and Telangana). Moran's- I showed a positive yet poor spatial clustering in the COVID-19 IR over neighboring districts. Our regression analysis demonstrated that percent of 15–59 aged population, district population density, percent of the urban population, district-level testing ratio, and percent of stunted children were significantly and positively associated with the COVID-19 infection ratio. We also found that, with an increasing percentage of literacy, there is a lower infection ratio in Indian districts.

**Competing interests:** The authors have declared that no competing interests exist.

## Conclusion

The COVID-19 infection ratio was found to be more rampant in districts with a higher working-age population, higher population density, a higher urban population, a higher testing ratio, and a higher level of stunted children. The study findings provide crucial information for policy discourse, emphasizing the vulnerability of the highly urbanized and densely populated areas.

## Introduction

Coronavirus disease 19 (COVID-19) is a respiratory disease caused by the SARS-CoV-2 virus, which is a member of a large family of viruses called coronaviruses. These viruses can infect people and some animals. The virus is thought to spread from person to person through droplets released when an infected person coughs, sneezes, or talks [1]. With more than 81,82,676 confirmed cases on October 31, 2020, India ranked second globally in terms of the total number of infected patients of COVID-19 [2]. The rate of spread of the disease was slow in the initial three months (January to March 2020) from the first outbreak in Kerala in January 2020, possibly because of the early nationwide lockdown [3–5]; widespread coverage about the pandemic in print, electronic and social media [6], and targeted efforts by the union and state governments on quarantine facilities and travel protocols [7,8]. There was a paid increase in the number of confirmed COVID -19 cases in many districts since April 2020. India has recorded over 50,000 cases every day from August 2020. S1 Fig shows that the trend of COVID-19 confirmed cases bi-weekly from March 14- October 31, 2020. After reaching a peak of COVID-19 infection in September- October 2020, new cases have been steadily declining in India. This biweekly peak reduced from 12,44,430 (12–25 September, 2020) to 5, 97,281 (24 October- 5 November, 2020).

Interestingly, India has a relatively high recovery rate and the lowest fatality rate globally [9]. Despite India's advantage of having a young age structure with less susceptibility to COVID-19 related deaths [10], India may have to undergo a higher burden of disease due to other demographic factors [11] such as the enormous population size, high population density, higher percentage of people living in poverty, lower levels of per capita public health infrastructure, and a high prevalence of co-morbid situations [12]. Research evidence that the transmission of second wave of COVID- 19 increase risk almost double of the first peak [13]. There is various factor associated with second wave more devasting such as–transmission dynamics, effect of population density, effect of testing rate and healthcare infrastructure [14].

Like any other health and demographic indicator, COVID-19 infection varies widely among the different states of the country [15,16]. However, the geographical pattern of the COVID-19 infection rate' does not coincide with the patterns of demographic and health indicators such as the under-five mortality rate or nutritional status. COVID-19 has been spreading rapidly in the urban areas, especially in states with megacities with densely populated urban slums like Delhi, Maharashtra, Tamil Nadu, and West Bengal. The sudden surge of return labour migration to the states of origin (due to COVID-19 related national lockdown), state-level health care system, adherence to physical distancing measures, and local government management are other potential community-level factors affecting geographical variations in the spread of COVID-19 in India. Some recent studies have computed composite indices to rank the districts in terms of their COVID-19 vulnerabilities using demographic information and infrastructure characteristics [17–19]. While such analyses help district-level planning and prioritization, they are based on the assumption that vulnerability will decrease as the districts' socioeconomic indicators improve. However, such an inverse relationship may

not be applicable in the context of COVID-19; for instance, a higher percentage of the urban population may indicate a higher socioeconomic status of the district population in a non-COVID situation and may be linked with an improved health outcome. However, it may be positively correlated with the spread of COVID-19. COVID-19 is more prevalent in cities and towns than in rural areas or hilly regions [20]. Therefore, it is imperative to unfold the empirical relationship patterns between the district's socioeconomic and household infrastructural characteristics and the COVID-19 infection ratio.

To the best of our knowledge, no such previous study has been conducted on COVID-19 in India. The aim of the present study is of two-folds, first to investigate the district level variation in COVID-19 during March-October 2020 and, secondly, to investigate the district level socio-economic and demographic correlates of COVID-19 infection ratio in India. Identification of such correlates is crucial for framing health policy and appropriate intervention.

## Data and methods

We used crowdsourced district-level data on COVID-19 available in the public domain from March 14, 2020, to October 31, 2020, accessed from the COVID-19 India dashboard. The time-series data on COVID-19 was available in the COVID-19 India from March 14, 2020 [21]. It is an application programming interface (API) to monitor the COVID-19 cases at national, state, and district levels. The data compiled in this web portal is based on state bulletins and official handles. The details of the data are available on the website. This portal data is consistent with the Ministry of Health and Family Welfare data, Government of India (https://www.mohfw.gov.in/) [22].

For explanatory variables, we utilized data from the National Family Health Survey of India 2015–16 (NFHS-4), a cross-sectional survey of 601,599 households, and 2.87 million individuals from all 29 states and seven union territories [23]. The survey collected data on various socioeconomic, demographic, health, and family planning indicators and anthropometry and biomarkers' measures related to anemia, hypertension, and diabetes. The NFHS-4 is the most recent source of such biomarker-based data at the district level in India. We also used some socioeconomic and demographic variables from the Census of India [24].

## Outcome variable

We have analyzed new, infected, recovered, and deceased cases at the national and state levels. The term "new cases" indicates the newly infected cases in the reference period; the term "confirmed cases" indicates the number of confirmed COVID-19 cases in the reference period. The recovered and deceased cases indicate the number of persons recovered and died from COVID-19 in the reference period. Finally, the term "average infected cases" means the total number of confirmed cases after excluding recovered and death cases.

For all the 640 districts in the thirty-five states and eight union territories of India, we defined the outcome variable, COVID-19 Infection Ratio (IR), as the number of confirmed cases in a given district per 100,000 population. For the district-level population for the year 2020, we projected the district population using an exponential growth rate from the census 2001 and 2011.

The infection of ratio was calculated as:

$$Infection\ Ratio\ (IR) = \frac{C_i}{P_i} * 100,000$$

Where,

$C_i$ = the number of confirmed cases in i[th] district and *the $P_i$* = total projected population in the i[th] district on October 31, 2020.

## District level correlates

Based on previous literature [25–28], we considered a set of 23 variables at district level, viz., i) Demographic variables: percentage of population aged 60 and above, percentage of population in age group 15–59, percentage of marginal worker, population density, ii) Socioeconomic variables: percentage of the literate population, percentage of Scheduled Castes (SC) population, percentage of Scheduled Tribes (ST) population, percentage of Hindu population(as an indicator of religious composition at district level), percentage of urban population, average number of persons that sleep in one room; iii) Household Infrastructure variables: percentage of households with availability of soap, percentage of households with water and toilet facility within the premise; iv) Health-related variables: percentage of women with Diabetes (Glucose>140mg), percentage of women (among age18+) with Cancer, percentage of 18+ aged women consuming tobacco, testing ratio per one hundred thousand population, under-five mortality rate, percentage of institutional births, percentage of full immunization among children aged 23–36 months, percentage of women aged 18 and above reporting anemia and, percentage of children with stunting and wasting. The district's health-related variables represent the district population's overall health status at the macro level.

## Statistical analysis

We performed a bi-weekly trend analysis of COVID-19 cases in India. To examine the district level correlates of the outcome variable, we carried out a linear regression analysis at the district level. Four separate district-level regression models were fitted—Model 1and Model 3 presents the independent variable's unadjusted effect without controlling any other independent variable's role. Model 2 and Model 4 show the adjusted results of the independent variables on the dependent variable. We did all the analyses in the statistical package Stata14.1. We tested for the possible multicollinearity among the independent variables before fitting them to the regression model.

## Spatial analysis

For analysing the spatial distribution of the COVID-19 cases at the district level, we generated descriptive maps of 640 districts in the software package QGIS. We later exported the shapefiles to GeoDa software to perform spatial analysis. Using the first-order 'Queen's contiguity matrix as the weight, we estimated Moran's I and univariate Local Indicators of Spatial Association (LISA). 'Moran's I" is the Pearson coefficient measure of spatial autocorrelation, which measures the degree to which data points are similar or dissimilar to their spatial neighbours [29]. The LISA cluster map yields four types of geographical clustering of the interest variable [30].

Here, "high-high" refers to the regions with an above-average infection ratio and sharing the boundaries with neighbouring areas with above-average infection ratio values. On the other hand, "high-low" indicates regions with below-average value and the surrounding areas with an above-average infection ratio. Also, the areas with below-average infection rates and sharing boundaries with neighbouring regions having values below the average of the same variables are referred to as low-low. The "high-high" are also referred to as *hot spots*, whereas the "low-low" is referred to as *cold spots* [30,31].

## Results

India has been reporting new cases of the coronavirus (COVID-19) every day since March 14, 2020. India reported over 8.1 million confirmed cases as of October 31, 2020. Out of these, around 7.4 million patients recovered, while 1 22,154 were fatal [32].

Fig 1 and S1 Table present the national bi-weekly (14 days) national pattern of new confirmed, infected, recovered, and deceased COVID-19 cases in India in the study period. In India, the average bi-weekly new confirmed cases rose 677 times (from 63 to 42,663), the average recovered points increased 10,729 times (from 5 to 53,647), the average infected patients increased 449 times (from 63 to 28,256). The average deceased cases increased 502 times (from 1 to 502) between the 1st and the 17h bi-weekly. During the study period, the COVID-19 cases were in peak 12th-25th September 2020, and then it started declining.

S2 Table presents national and state-wise confirmed, infected, recovered, and deceased cases for the entire study period (March 14, 2020, to October 31, 2020). It also presented

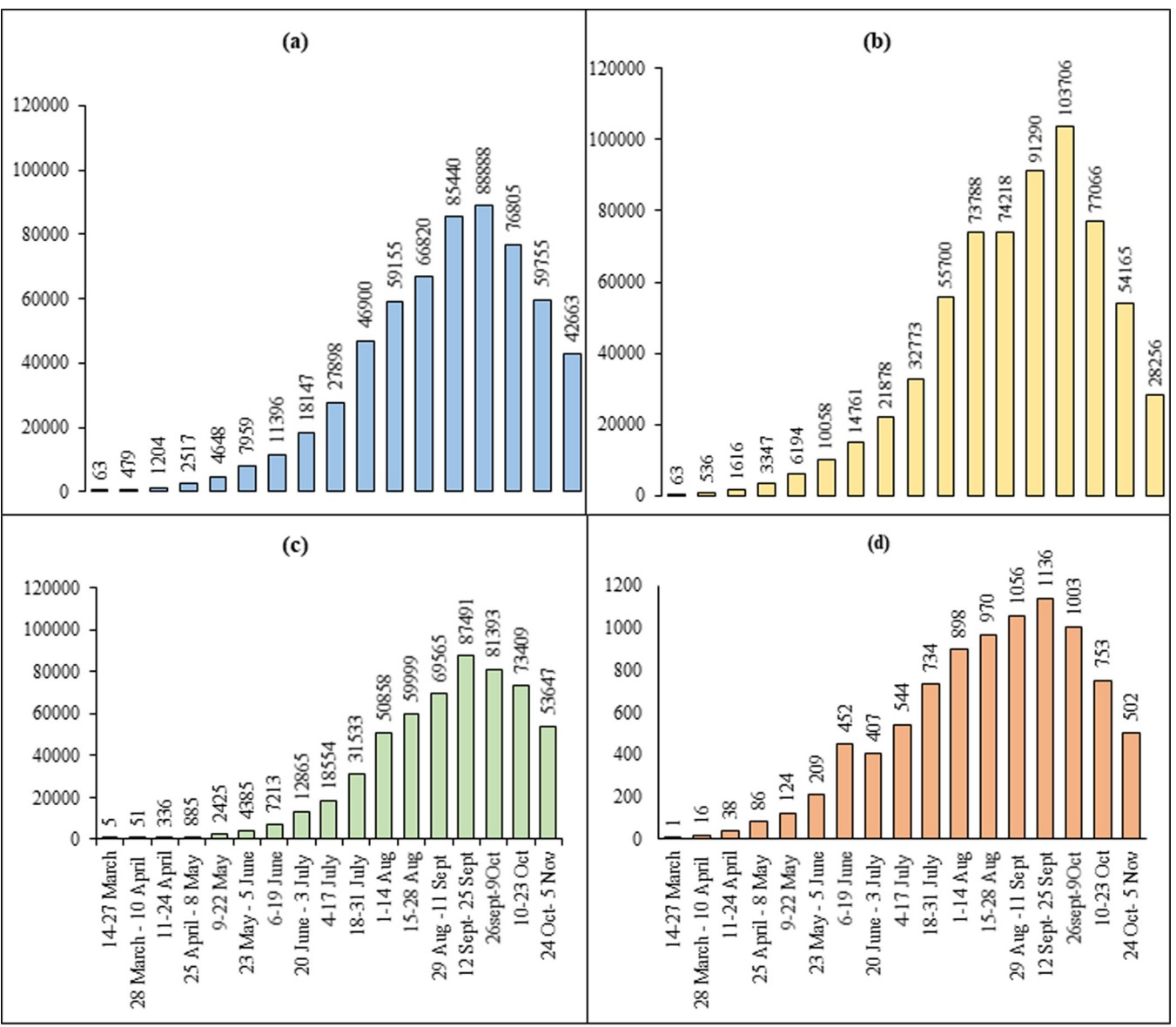

Source: Author's calculations

**Fig 1. Bi-weekly average new, infected, recovered, deceased cases in India (March 14 –November 5, 2020).** (a). Bi-weekly average new cases; (b). Bi-weekly average infected cases bi-weekly; (c). Bi-weekly average recovered cases; (d). Bi-weekly average deceased cases.

infection ratios at the national and state level. It is found that six states viz. Maharashtra, Andhra Pradesh, Tamil Nadu, Karnataka, Telangana, and Uttar Pradesh, have accounted for more than half of the country's total cases. About 70 percent of new possibilities, 70 percent of recovered cases, 80 percent of infected patients, and 61 percent of COVID-19 deaths are from only nine states (Maharashtra, Delhi, Tamil Nadu, Karnataka, Andhra Pradesh, Telangana, West Bengal, Gujarat, and Uttar Pradesh). The IR in India is 42.85 per one hundred thousand people. The highest IR was observed in Kerala (259.93) and the lowest in Bihar (6.58). However, the states like Arunachal Pradesh (123.40), Delhi (165.13), Goa (152.21), Ladakh (223.89), Maharashtra (101.63), Manipur (121.58), and Puducherry (245.95) experienced a far higher level of infection ratio than the national average. Only two Union Territories, viz., Lakshadweep, and Daman & Diu, experienced zero cases during the study period.

### District level variations in COVID-19

**Panel A in Figs 2 and 3** showed the district-level variations in COVID-19 on July 31, 2020, and October 31, 2020, respectively. The size of the bubble in the figure indicates the number of positive COVID-19 cases in districts on these two time periods. The larger the size of the bubble, the higher is the number of confirmed cases. Of all districts, a shift in the number of cases could easily visible in districts ranking till July 31, 2020. Mumbai has the highest number of cases, while by the end of October 2020, Bengaluru Urban tops the confirmed cases. By the end of July 2020, five urban districts contain about 28 percent of the confirmed cases (Mumbai, 7.02 percent; Chennai, 6.13 percent; Thane, 5.73 percent; Pune, 5.48 percent and Bengaluru, 3.41 percent). However, a new pattern has been observed on October 31, 2020, when about 17 percent of the COVID-19 confirmed cases consists of five major districts (Bengaluru Urban: 4.25 percent; Pune: 4.22 percent; Mumbai:3.25 percent; Thane: 2.82 percent and Chennai: 2.52 percent). In July 2020, about 11 percent of the confirmed cases belonged to another seven urban districts (Hyderabad, Central Delhi, Ahmedabad, South East Delhi, Kolkata, West Delhi, and East Godavari) with at least 20,000 confirmed cases. However, by the end of October 31, 2020, seven districts contributed 9.34 percent of total cases, with at least 90,000 confirmed cases in each district. About 90 percent (575 districts) have at least 1000 confirmed cases each, and about 97 percent (625 districts) have at least 100 positive confirmed cases of COVID-19. About 99.5 percent of the districts reported at least one confirmed case of COVID-19 until October 31, 2020.

In Figs 2 and 3, **panel B** presents the district level infection ratio (IR), defined as the number of confirmed cases per 100,000 population on July 31, 2020, and October 31, 2020, respectively. The results show that the top three worst-affected districts (Central Delhi, New Delhi, Mumbai, and Chennai) in India have at least 2,000 (per 100,000 population). Another three districts, viz., South East Delhi, Kamrup Metropolitan, and North Delhi, have IR ranging between 1000 to 2000 per one hundred thousand population till the end of July 2020. However, by the October end, the same districts have experienced a manifold increase in IR (for instance, the IR in Kamrup Metropolitan increased from 1,035 to 77,044 per 100,000 population). On October 31, 2020, East Delhi, Kamrup Metropolitan, Jodhpur, Ranga Reddy, and Ajmer were the top five worst districts experiencing IR from 26,456 to 96,256 per one hundred thousand population. There were about eleven districts with an IR ranging between 11,780 and 24,806 per one hundred thousand people. These were Kolkata, Warangal, Central Delhi, Bengaluru Urban, Kannur, Kurnool, Fatehabad, Udupi, Madhubani, West Delhi Haridwar by October 31, 2020.

**Fig 4** presented the Moran's I and LISA cluster maps for the district infection ratio of COVID-19 in India for March 14, 2020- July 31, 2020, and March 14, 2020- October 31, 2020.

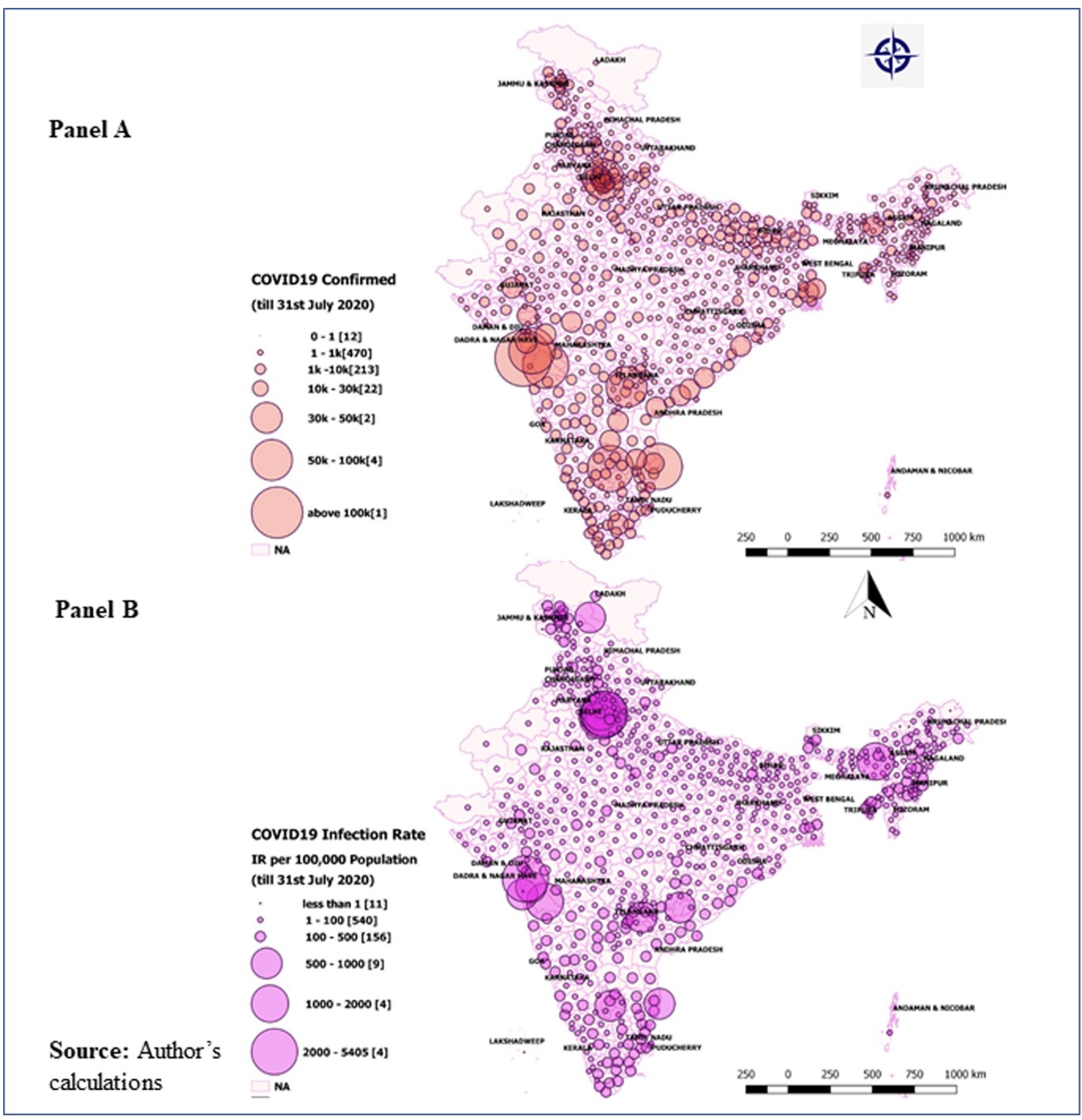

**Fig 2. District-level variations in COVID-19 till July 31, 2020.** Panel A: Absolute number of COVID-19 cases in districts as of July 31, 2020; Panel B: District level infection ratio (IR), defined as the number of confirmed cases per 100,000 population by July 31, 2020.

Both maps represented a positive spatial clustering level ('Moran's I value are 0.333 and 0.282 respectively) in the COVID-19 infection rate over neighbouring districts. Hotspots of COVID-19 were observed in the parts of Konkan coast of Maharashtra (Palghar, Mumbai, Thane, Nashik, and Satara); in the southern part from Tamil Nadu (Chennai, Chengalpattu,

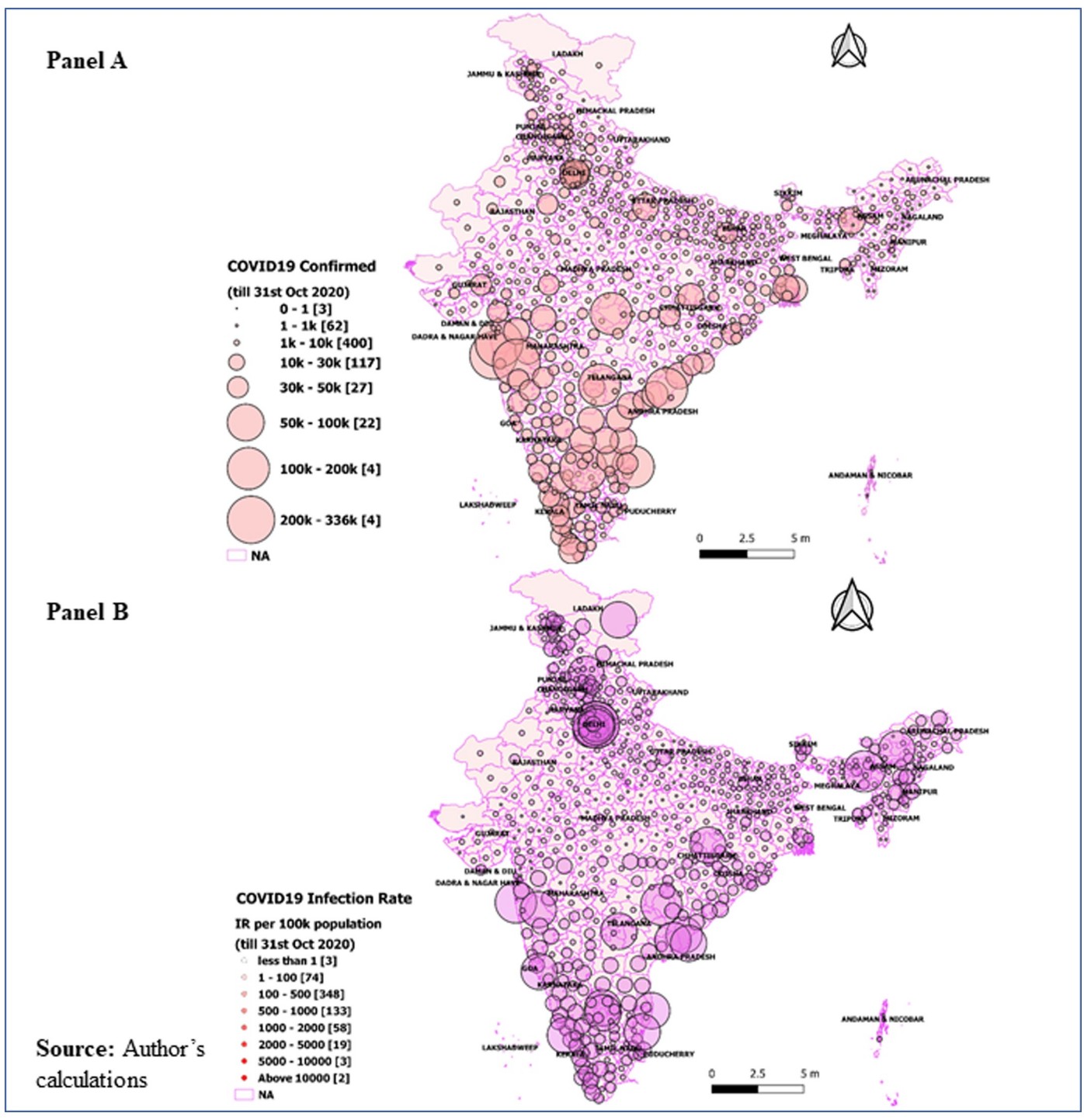

**Fig 3. District-level variations in COVID-19 till October 31, 2020.** Panel A: Absolute number of COVID-19 cases in districts till October 31, 2020; Panel B: District level infection ratio (IR), defined as the number of confirmed cases per 100,000 population by October 31, 2020.

Thiruvallur, Virudhunagar, and Ramanathapuram); in Delhi; and in the northern part of Jammu & Kashmir (Ganderbal and Pulwama) whereas cold-spots were observed in central, north-western and north-eastern regions of India.

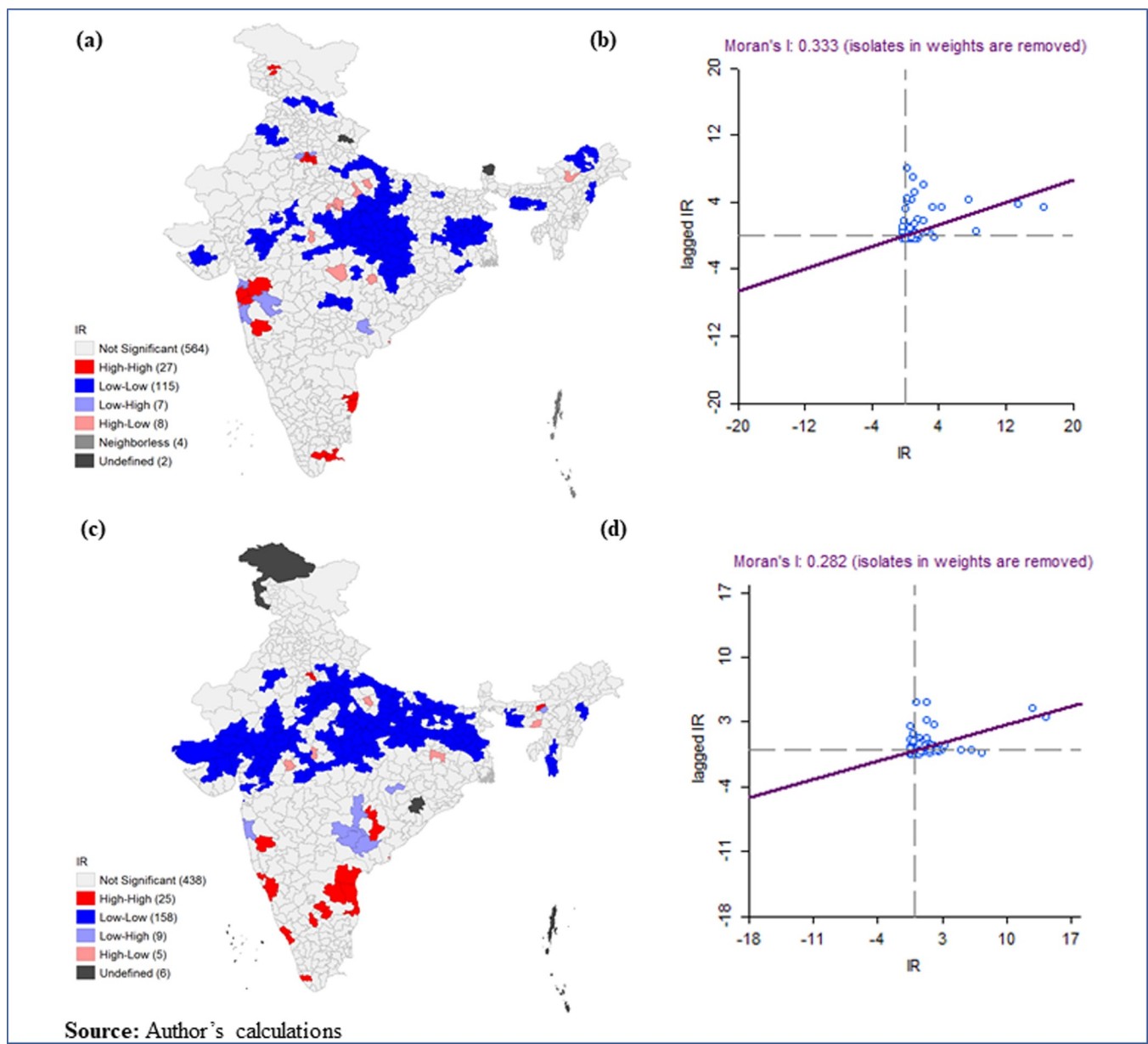

**Fig 4. Moran's I and LISA cluster maps for district infection ratio of COVID-19 in India for July 31, 2020 and October 31, 2020.** (a). LISA cluster map, July 31, 2020; (b). Moran's I, July 31, 2020; (c). LISA cluster map October 31, 2020; (d). Moran's I, October 31, 2020.

**Table 1** shows the descriptive statistics of the outcome variable (Infection Ratio) and exposure variables for the 640 districts. The average infection ratio is (108.40 per one hundred thousand population on July 31, 2020, and 586.54 per one hundred thousand population on October 31, 2020. On July 31, 2020, the IR was zero in the two districts from Arunachal Pradesh (Dibang valley, Krumung Kumey) and another two districts from the Union Territories (Lakshadweep and Nicobar).

All district-level socioeconomic variables differ substantially among districts. For example, the mean percent of the old age population (60 and above) is 8.33, and it varies between 2.46 percent to 17.82 percent among the districts of India. The percent of marginal worker a proxy

**Table 1. Summary statistics for the outcome and explanatory variables across 640 districts of India.**

| Indicator | Mean | SD. | Min | Max |
|---|---|---|---|---|
| Infection Ratio per 100,000 population (March 14 July 31, 2020) | 108.40 | 374.146 | 0.00 | 5918.43 |
| **Infection Ratio per 100,000 population** (March 14 31 October 2020) | 586.54 | 1149.29 | 0.00 | 16977.18 |
| **Demographic variables** | | | | |
| Population working 15–59** | | | | |
| Percent of 60 years and above population** | 8.33 | 2.05 | 2.46 | 17.82 |
| **Percent of marginal workers**** | 10.97 | 5.63 | 1.36 | 33.60 |
| Population density | 936.18 | 3053.32 | 1.00 | 36155.00 |
| **Socioeconomic variables** | | | | |
| Percent of literate population | 62.46 | 10.53 | 28.77 | 88.74 |
| Percent of SC population ** | 14.86 | 9.13 | 0.00 | 50.17 |
| Percent of ST population** | 17.71 | 27.00 | 0.00 | 98.58 |
| Percent of Hindu population** | 74.00 | 26.79 | 0.85 | 99.39 |
| Percent of Urban population | 26.40 | 21.12 | 0.00 | 100.00 |
| The average number of persons sleeping in a room | 2.92 | 0.55 | 1.60 | 4.40 |
| **Household Infrastructure variables** | | | | |
| Percent of households with soap availability | 62.20 | 19.58 | 17.20 | 98.90 |
| Percent of households with water availability within the premise | 48.14 | 26.59 | 4.20 | 100.00 |
| Percent of households with Toilet facility within the premise | 61.41 | 27.73 | 10.10 | 100.00 |
| **Health-related variables** | | | | |
| Percent of women with Diabetes (Glucose>140mg) | 6.02 | 2.22 | 1.10 | 13.70 |
| Percent of women (among age18+) who reported having Cancer Disease | 0.15 | 0.45 | 0.00 | 6.5 |
| Percent of 18+aged women consuming Tobacco | 10.52 | 13.22 | 0.00 | 81.1 |
| Testing ratio per 100 000 | 2167.57 | 2142.60 | 0.00 | 11471.91 |
| Under-5 mortality rate | 46.58 | 22.96 | 2.01 | 128.63 |
| Percent of institutional births | 78.97 | 17.31 | 9.74 | 100.00 |
| Percent of full immunization among children aged 23–36 months | 62.40 | 17.42 | 7.14 | 100.00 |
| Percent of women reporting anemia among women aged18+ | 51.53 | 12.24 | 13.06 | 84.25 |
| Percent of children with stunting | 35.96 | 9.92 | 12.41 | 65.11 |
| Percent of children with wasting | 20.59 | 7.66 | 1.77 | 46.93 |

Source: *Infection Ratio (IR) was computed from COVID- 19 Dashboard of India.

** Variables were computed from Census of India 2011; The rest of the explanatory variables were calculated from the fourth round of the National Family Health Survey [23].

indicator of district level out migration, ranged substantially from 1.36 to 33.60 percent. Population density (number of persons per square kilometer) varies from 1 in Dibang Valley (Arunachal Pradesh) to 36155in northeast Delhi.

We also observed massive variation in the districts' socioeconomic variables, viz., percent Hindu, urban, and ST population vary from 0 percent to 100 percent in India's districts. On average, 2.92 persons sleep in one room in the Indian districts. On average, 48.14 percent of the households have water facilities, and 61.41 percent have toilet facilities within the household premises. Hygiene practice of availability of soap ranges from less than 17.20 percent to 98.90 percent in India. There exists a wide disparity in health-related variables too. While the testing ratio ranged from 0 to11471 persons per 100 hundred thousand, the percentage of full immunization among children ranged from 7.14 to 100. The tobacco consumption among women ranged from 0.8 to 88 percent.

**Table 2. Regression analysis of district correlates and infection ratio (IR) across 640 districts of India[1].**

| Models | 31-Jul-20 | | | | | | 31-Oct-20 | | | | | |
|---|---|---|---|---|---|---|---|---|---|---|---|---|
| | Unadjusted model (1) | | | Adjusted model (2) | | | Unadjusted model (3) | | | Adjusted model (4) | | |
| | β-Coeff. | p-value | 95% CI | β-Coeff. | p-value | 95% CI | β-Coeff. | p-value | 95% CI | β-Coeff. | p-value | 95% CI |
| **Demographic variables** | | | | | | | | | | | | |
| Percent of 15–59 aged population | 21.29 | 0.000 | (15.31,27.27) | 16.91 | 0.000 | (5.27,28.55) | 90.38 | 0.000 | (72.51,108.25) | 75.73 | 0.000 | (39.71,111.74) |
| percent of 60 years and above population | -1.42 | 0.840 | (-15.58,12.74) | -6 | 0.570 | (-26.93,14.93) | 39.12 | 0.080 | (-4.28,82.52) | 39.6 | 0.220 | (-23.85,103.04) |
| Percent of marginal workers | -13.08 | 0.000 | (-18.15,-8.01) | 3.13 | 0.350 | (-3.44,9.69) | -41.54 | 0.000 | (-57.16,-25.93) | 15.7 | 0.130 | (-4.67,36.07) |
| Population density | 0.06 | 0.000 | (0.06,0.07) | 0.05 | 0.000 | (0.04,0.06) | 0.18 | 0.000 | (0.15,0.2) | 0.12 | 0.000 | (0.09,0.15) |
| **Socio-economic variables** | | | | | | | | | | | | |
| Percent of literate population | 7.15 | 0.000 | (4.44,9.85) | -0.54 | 0.830 | (-5.46,4.38) | 29.39 | 0.000 | (21.17,37.61) | -12.84 | 0.090 | (-27.86,2.18) |
| Percent of SC population | 0.43 | 0.790 | (-2.76,3.62) | 2.93 | 0.130 | (-0.9,6.77) | -4.13 | 0.410 | (-13.96,5.7) | 3.33 | 0.590 | (-8.67,15.34) |
| Percent of ST population | -1.22 | 0.030 | (-2.29,-0.14) | -0.27 | 0.750 | (-1.98,1.44) | -2.53 | 0.140 | (-5.86,0.8) | 2.98 | 0.270 | (-2.34,8.31) |
| Percent of Hindu population | -0.09 | 0.870 | (-1.18,1) | -0.24 | 0.760 | (-1.81,1.32) | -0.12 | 0.940 | (-3.49,3.24) | 3.05 | 0.220 | (-1.8,7.89) |
| Percent of Urban population | 7.27 | 0.000 | (6.01,8.53) | 2.45 | 0.020 | (0.39,4.51) | 24.63 | 0.000 | (20.79,28.48) | 12.71 | 0.000 | (6.37,19.04) |
| Average number of persons sleeping in a room | -28.21 | 0.300 | (-81.04,24.62) | 10.95 | 0.780 | (-66.95,88.85) | -341.62 | 0.000 | (-502.28,-180.96) | -86.52 | 0.480 | (-327,153.95) |
| **Household infrastructure variables** | | | | | | | | | | | | |
| Percent of households with soap availability | 3.5 | 0.000 | (2.04,4.96) | 0.38 | 0.740 | (-1.84,2.59) | 11.72 | 0.000 | (7.24,16.2) | 4.01 | 0.250 | (-2.76,10.78) |
| Percent of households with water availability within the premise | -0.17 | 0.750 | (-1.27,0.92) | -0.28 | 0.640 | (-1.46,0.9) | 0.11 | 0.950 | (-3.27,3.48) | 2.09 | 0.260 | (-1.52,5.7) |
| Percent of households with Toilet facility within the premise | 2.41 | 0.000 | (1.38,3.45) | -0.14 | 0.900 | (-2.31,2.04) | 9.61 | 0.000 | (6.47,12.75) | -2.33 | 0.490 | (-9.01,4.35) |
| **Health-related variables** | | | | | | | | | | | | |
| Percent of women with Diabetes (Glucose>140mg) | 21.05 | 0.000 | (8.06,34.05) | 0.41 | 0.950 | (-13.2,14.03) | 89.86 | 0.000 | (49.94,129.78) | -5.6 | 0.800 | (-47.94,36.74) |
| Percent of women (among age18+) who reported having Cancer Disease | 25.6 | 0.440 | (-39.49,90.68) | 29.19 | 0.300 | (-26.53,84.9) | 28.47 | 0.780 | (-171.57,228.51) | 39.5 | 0.650 | (-130.98,209.98) |
| Percent of 18+aged women consuming Tobacco | -2.68 | 0.020 | (-4.87,-0.49) | 0.24 | 0.860 | (-2.36,2.84) | -8.54 | 0.020 | (-15.39,-1.68) | 1.9 | 0.650 | (-6.31,10.1) |
| Testing ratio per 100000 | 0.03 | 0.000 | (0.02,0.04) | 0.03 | 0.000 | (0.01,0.04) | 0.04 | 0.000 | (0.02,0.05) | 0.03 | 0.000 | (0.02,0.04) |
| Under-5 mortality rate | -2.59 | 0.000 | (-3.84, -1.33) | -0.99 | 0.210 | (-2.54,0.55) | -12.59 | 0.000 | (-16.37,-8.82) | -3.7 | 0.130 | (-8.44,1.05) |
| Percent of institutional births | 2.94 | 0.000 | (1.28,4.6) | -0.72 | 0.570 | (-3.24,1.79) | 13.36 | 0.000 | (8.29,18.43) | -4.12 | 0.290 | (-11.79,3.56) |
| Percent of full immunization among children aged 23–36 months | 0.81 | 0.340 | (-0.86,2.48) | -1.27 | 0.170 | (-3.07,0.53) | 8.32 | 0.000 | (3.22,13.42) | -0.62 | 0.830 | (-6.16,4.92) |
| Percent of women reporting anemia among women aged18+ | 0.39 | 0.750 | (-1.98,2.77) | 0.18 | 0.890 | (-2.29,2.64) | -2.55 | 0.490 | (-9.85,4.75) | -3.01 | 0.430 | (-10.56,4.55) |
| Percent of children with stunting | -4.58 | 0.000 | (-7.49,-1.67) | 4.46 | 0.050 | (-0.04,8.96) | -25.28 | 0.000 | (-34.11,-16.46) | 12.12 | 0.080 | (-1.62,25.86) |
| Percent of children with wasting | -1.17 | 0.550 | (-4.97,2.63) | 1.88 | 0.380 | (-2.31,6.06) | -5.74 | 0.340 | (-17.43,5.96) | 4.29 | 0.510 | (-8.53,17.12) |

Source: Author's Computation.

[1] All variables are computed at the district level.

The regression analysis of infection ratio (presented in **Table 2**) displays both unadjusted and adjusted coefficients of the exploratory variables for two time periods, viz., March 14, 2020-July 31, 2020 and March 14, 2020-October 31, 2020. In the unadjusted models (1 & 3), among demographic variables, percent of 15–59 aged population, percent of marginal workers, and population density were significantly associated with IRs. Several socioeconomic variables, such as the literate population, ST population, and urban population, and the average person sleeping in a room were significantly associated in the unadjusted model 1. In unadjusted model 3, percent of urban population is significantly associated with COVID-19 infection together with the percent of literate population, and the average person sleeping in a room Also, districts with better household infrastructure facilities have a higher likelihood of COVID-19 infection IR.

Districts with higher levels of household infrastructure facilities (such as soap or toilet facility) reported higher levels of IRs in model 3. Among the health-related variables, percentage of women with diabetes (glucose > 140), testing ratio, institutional births, percentage of children immunized were significantly positively correlated with COVID-19 IR in October in the unadjusted model. Districts with high diabetic patients have a 21.05 and 89.8690-fold higher chance of COVID-19 infection in July and October than those with a low prevalence of diabetes. However, the percentage of women consuming tobacco, under-5 mortality, and children with stunting conditions were associated negatively with COVID-19 IR.

In the adjusted model (2 & 4), the association between IRs and most of the correlates becomes statistically insignificant. After controlling the roles of demographic, socio-economic, household facilities, and health-related variables in the adjusted model, percent of 15–59 aged population (model 2: 16.91, CI:5.27–28.55 vs model 4: 75.33, CI:39.71–111.74), district-level population density (model 2: 0.05, CI:0.04–0.06 vs model 4: 0.12, CI:0.09–0.15), and percent of urban population (model 2: 2.45, CI: 0.39, 4.51 vs model 4: 12.7, CI: 6.37–19.04).) were positively significantly associated with the district infection ratio at district level. Thus, with an increasing percentage level of these variables at the district level, there was an increasing chance of COVID-19 infection ratio. As the districts' literacy rate increases, the COVID-19 infection ratio also decreases in the adjusted models, but the relationship is statistically significant only in the infection ratio until October 31, 2020 (model 4: -12.84, CI:27.86, 2.18).

In the adjusted models, none of the infrastructure variables were significantly related to the spread of virus. However, among the health-related variables the district-level, testing ratio (model 2: 0.03, CI: 0.01–0.04 vs. model 4: 0.03, CI: 0.02–0.04) and child stunting (model 2: 4.46, CI: -0.04–8.96 vs model 4: 12.12, CI: -1.62–25.86) was significantly and positively associated with the prevalence of effective in containing the COVID-19 virus.

## Summary and discussion

In terms of the total number of confirmed cases, India ranked second after the US, reporting more than eight million COVID-19 cases as of October 31, 2020. The present study examined the district-level variation in COVID-19 cases from March 14, 2020, to October 31, 2020. The present study also aimed to identify socioeconomic and demographic correlates of COVID-19 infection ratio at the district level. Due to different stages of socio-economic development in the states, the trajectory of COVID-19 and related intervention cannot be uniform. Our result illustrated the differences in COVID-19 cases at the state and district levels with few critical findings.

First, the spread of COVID-19 has been increasing over time during the study period. The average bi-weekly cases show that the new, infected, recovered, and deceased cases are growing nationally. It is found that about 80 percent of the infected patient and 61 percent of the deaths

are concentrated in nine states (Delhi, Gujarat, West Bengal, Uttar Pradesh, Andhra Pradesh, Maharashtra, Karnataka, Tamil Nadu, and Telangana). On October 31, 2020, the IR in India was 42.85 per hundred thousand population. The IR ranged from a minimum of 6.58 in Bihar to a maximum of 259.63 in Kerala. Only two Union Territories (Lakshadweep and Daman & Diu) have zero IR. The metropolitan cities like New Delhi, Mumbai, Thane, Pune, Kamrup Metropolitan, South Goa, Chennai, Bengaluru, and Hyderabad were most affected by COVID-19. The study identifies the districts at higher risk of coronavirus infection in the southern, northern, and western states. The apparent concern is that these states also contribute significantly to the Indian economy [33]. Another important observation of this study is that districts bordering the six metropolitan cities were observed to be India's highest hot spots, possibly because they contribute the largest share of migrants and commuters to these megacities.

Spatial autocorrelation analysis of COVID-19 infection ratio shows that a district's infection ratio is not highly correlated with that of the neighboring districts. We have observed a few hotspots of COVID-19 in Maharashtra, Tamil Nadu, Delhi, and Jammu &Kashmir. In contrast, we identified cold-spots in the central, north-western and north-eastern regions of India.

Finally, our research reveals that the district's infection ratio of COVID-19 is correlated with most socioeconomic and health-related variables. However, after adjusting other variables' roles, we observed a statistically significant association only with a limited number of variables. After adjusting the role of socioeconomic and health-related factors, the COVID-19 infection ratio was found to be higher in districts with a higher working-age population, higher population density, a higher urban population, a higher testing ratio, and a higher level of stunted children.

The results obtained in regression analysis are consistent with that from the geographical analysis of the Covid-19 infection ratio. It has been observed that highly urbanized districts are worst affected by Covid-19. Population density is also higher in urban areas compared to rural areas. As the percentage of the urban population increases, the chances of unavoidable economic activities such as medicine, food transport, distribution, etc., also increase even in the national lockdown period, which exposes more people to the pandemic. Previous studies also showed that higher population densities in congested slum areas and large towns accelerated COVID-19 infection and mortality rates [34–36]. The congestion, slum concentrations, inadequate housing, and sanitation in poor urban areas may explain such high disease. A positive association between COVID-19, IRs, and testing ratio indicates underreporting of COVID-19 in districts where the testing ratio is low.

Studies based on individual data show that older people are more vulnerable to COVID-19 infections [37,38]. This study also identified that pre-existing diabetes is positively associated with COVID-19 disease [37,38]. In our research, we did not find such associations since our analysis is not an individual-level analysis. One of the major limitations of our study is not extending the analysis beyond October 31, 2020. However, this analysis can be extended in future. Yet, unlike many previous studies, we are identifying macro-level correlates of the COVID-19 infection rate for the period March 14, 2020, to October 31, 2020.

## Comparison between first and second waves of COVID-19 in India

At the onset of the COVID-19 pandemic, India imposed world's strictest nationwide lockdown beginning from March 25, 2020 [39], But, as of April 10, 2021, India was the third leading country after USA and Brazil's identified cases [40]. Like several other parts of the world, India has been experiencing a massive surge of COVID-19 cases and deaths [41–43]. The second wave has started in the middle of March, 2021 and recorded highest number of cases

(144,829) on April 09, 2021 [42,43]. The major affected states were Maharashtra, Kerala, Karnataka, Andhra Pradesh, Tamil Nadu, Andhra Pradesh, Delhi, Uttar Pradesh and West Bengal [39,41,42]. Moreover, several megacities with high concentration of population and overcrowded with migrants registered high transmission rate of the disease. Mumbai Urban and NCT of Delhi were the example of two such cities. Despite this high caseload, several national movements such as the farmer's protest in November 2020 at the New Delhi border, elections in several states and some religious gatherings have kept the social distancing norms at stake and made situation highly vulnerable for the spread of the virus. Another reason for the high spread of virus might be the intra-urban mobility form slum to towns/cities. The household density of the slum areas is one of the most vital causes of the infection spread. The international airports of NCT of Delhi, Mumbai, Bengaluru and Kolkata acted as the gateways to allow the dangerous mutating COVID-19 strain to spread across the main urban hubs of the country. Moreover, migrant labors also borrow the new strain to the different corners of the country. We also found that previous studies on second wave, have similar pattern of hotspots clusters with as our study. The concentration of the virus spread basically found in Mumbai Urban -Pune-Nasik-Kolhapur region and NCT of Delhi (comprising Punjab and Haryana) [39,41–44]. Though Kerala and districts surroundings Bengaluru Urban have high concentration of confirmed cases, it didn't report high CFR. These patterns would give us some idea to combat with the present situation and get prepared for the predicted third wave.

## Conclusion

The COVID-19 pandemic is expected to have a long-term impact on health, economy, and social processes globally, including India. Only a clear understanding of the disease's spatial distribution and its correlates will help to formulate policies and interventions. Therefore, the possible risk factors should be included in policy preparedness and implementation during the COVID-19 pandemic. Understanding risk factors of COVID-19 may also help to understand the future dynamics of COVID-19 or other such infectious diseases.

We found that the share of working-age populations, population density, urban residence, and testing rates are significantly correlated with the COVID-19 infection ratio (IR). As in urban areas, the population density is very high, and social distancing is challenging to maintain; the role of government is crucial in combating the pandemic. By ensuring the health and hygiene-related facilities, (providing adequate clean water, adequate sanitation, and sewerage facilities, cleaning the city, maintaining quarantine centers and public health care institutions, etc.), and improving public distribution system to ensure minimum food supply, especially among the urban poor and other deprived sub-groups, can help to control the spread of COVID-19 infection.

More tests are required to classify patients with asymptomatic conditions. India has a population of over 1.3 billion, but till October 31, 2020, approximately 1087.9 million (only 8.04 percent) tests have been carried out. As of April 1, 2021, India has 552,566 active infection cases while 11,434,301 patients have recovered and 162,468 have succumbed to COVID-19. According to Ministry of Health and Family Welfare, a total of about 640.05 million doses of COVID-19 vaccine have been administered by August 31, 2021 [45]. Simultaneously, people's negligent behavior towards COVID-19 protocols (like, not following the social distancing norms, not wearing the mask in public places, and coughing without covering mouth) put them at a higher risk. Finally, there is a need to improve infrastructure (hospitals, ventilators, PPE kits) and human resources (doctors, nurses, and frontline workers) in healthcare facilities.

Our analysis does have a few limitations. First, there is a possibility of under-reporting positive and fatal cases due to a lack of testing or social stigma. Hence our data gives the most

conservative estimates of the infection ratio. Second, for most cases, the patients' level of information (such as age, sex, and comorbidity) is unavailable. Therefore, we analyzed the district-level determinants instead of individual-level determinants. Thus, our results identified the major correlates only at the district level. Finally, we analyzed the number of confirmed cases for infection ratio rather than the number of active cases. The later considers the recovery rate and depends on the health service available in a region. We used the number of confirmed cases as the primary indicator of the spread of the infection. Despite these limitations, the study's merit lies in bringing together spatial-demographic vulnerabilities prevalent across the nation during the pandemic period.

## Supporting information

**S1 Fig. Trend of bi-weekly total confirmed cases in India, 14 March–November 5, 2020.**
Source: Author's calculations.
(DOCX)

**S1 Table. Average number of new cases, recovered, infected, deceased cases bi-weekly (14 days) in India (March 14, 2020 –November 5, 2020).** Source: Author's Computation.
(DOCX)

**S2 Table. State-wise COVID-19 cases in India (14th March - 31st October, 2020).** Source: Author's Computation. Note: Computation of Infected Cases $I_i = (C_i - R_i - D_i)$.
(DOCX)

**S3 Table. Brief description of each indicator used in this analysis.** Source: Infection Ratio (IR) from provided data from COVID- 19 Dashboard of India. Independent variables computed from Census of India (2011) and fourth round of National Family Health Survey (IIPS;2017); **mortality estimates (reference period five year of the preceding.
(DOCX)

## Author Contributions

**Conceptualization:** Vandana Tamrakar, Ankita Srivastava, Nandita Saikia.

**Data curation:** Sudheer Kumar Shukla.

**Formal analysis:** Vandana Tamrakar, Ankita Srivastava, Mukesh C. Parmar.

**Investigation:** Vandana Tamrakar, Ankita Srivastava.

**Methodology:** Vandana Tamrakar, Ankita Srivastava, Nandita Saikia.

**Supervision:** Nandita Saikia.

**Validation:** Nandita Saikia.

**Visualization:** Nandita Saikia.

**Writing – original draft:** Vandana Tamrakar, Ankita Srivastava, Nandita Saikia, Shewli Shabnam, Bandita Boro, Apala Saha, Benjamin Debbarma.

**Writing – review & editing:** Nandita Saikia.

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
