## [Decision Letter · Decision Letter 0]

18 Feb 2021

PONE-D-20-34338

District level correlates of COVID-19 pandemic in India

PLOS ONE

Dear Dr. Saikia,

Thank you for submitting your manuscript to PLOS ONE. After careful consideration, we feel that it has merit but does not fully meet PLOS ONE’s publication criteria as it currently stands. Therefore, we invite you to submit a revised version of the manuscript that addresses the points raised during the review process.

Considering the reviewer comments, I am going with a decision of major revision. Either your respond to reviewer comments or revise according to their suggestions. We would like to see a revised version of the paper. 

We look forward to receiving your revised manuscript.

Kind regards,

Srinivas Goli, Ph.D.

Academic Editor

PLOS ONE

Journal Requirements:

2. Please include a copy of Table A1 which you refer to in your text on page 5.

Additional Editor Comments:

Considering the reviewer comments, I am going with a decision of major revision. Either your respond to reviewer comments or revise according to their suggestions. We would like to see a revised version of the paper.

Reviewers' comments:

Reviewer's Responses to Questions

**Comments to the Author**

1. Is the manuscript technically sound, and do the data support the conclusions?

Reviewer #1: Partly

Reviewer #2: Partly

2. Has the statistical analysis been performed appropriately and rigorously? 

Reviewer #1: N/A

Reviewer #2: N/A

3. Have the authors made all data underlying the findings in their manuscript fully available?

Reviewer #1: Yes

Reviewer #2: Yes

4. Is the manuscript presented in an intelligible fashion and written in standard English?

Reviewer #1: Yes

Reviewer #2: Yes

5. Review Comments to the Author

Reviewer #1: Thank you for the opportunity to review this manuscript on a very topical issue. The overall idea, writing style and intent is good . The concerns regarding the study itself are however as below:

1. The study analyses data between march to July 2020 and therefore the timeliness of the publication and its findings itself may not resonate with the reader. Apart from the fact that things have changed since this study was done, the pandemic is now on the decline and in reality, widely prevalent in states like Kerala which did not need mention early in the pandemic given the State did so well.

2. It may be pertinent to revisit the timeline and consider maybe an analysis atleast till the start of the winter months to make this a more relevant piece today.

3. The use of open source data , including NFHS is a good step, but NFHS data from next round too is now available.

4. rather than only using above 60 age group, considering the middle age-groups - the working age groups may have also helped give important insights.

5. The Sociodemographic variables did not include one on household income or occupation, while education may have filled some of that gap but skewed the findings.

6. I was intrigued too by the selection of health indicators- fasting glucose , use of tobacco etc for women? Would this not under-represent prevalence across males of important co-variates like smoking? The access to healthcare is another important variable besides the consideration of pre-existing respiratory illness. A lot of selected variables related to immunisation and childhood illness and health indicators while this age group was largely the unaffected age group in the pandemic.

7.Line 283 - has an important spell edit - in terms of CoVID-19 protocols- not wearing the mask in pubic place to be edited to public place!

8. The overall intention of district-level correlates determining oVID-19 and feeding policy and management of the pandemic will be well-served if the right variables are selected for the analyses, and over a longer time window to help district-level planning and management.

9. Use of the infection ration is welcome but the correlates and determinants need addressing...

10. Thank you for the clear and succinct writing style but do revisit some of the points addressed above.

Reviewer #2: The study on “District level correlates of COVID-19 pandemic in India” about exploring the hot and cold spots in terms of districts and finding the socio-economic, demographic and health-related determinants of COVID-19 disease in India. The study uses publicly available data from COVID19 India API. The study uses descriptive statistics of COVID-19 disease and linear regression model using indicators from NFHS-4 and Census of India to establish the findings. The outcomes of the study are based on three major findings from: (a) trend analysis of COVID-19 cases, (b) infection ratio (hot and cold spots) and Moran’s Index and (c) determinants based on regression.

The study needs to give attention to these corrections:

Main comments:

1) The Introduction section, Summary and Discussion, and Conclusion section need to be strengthened. Introduction section may be well connected with the other sections of the manuscript.

2) Trend analysis, infection ratio, hot and cold spots, and regression results are interpreted independently. Trend analysis may be interpreted for its plausible role of the concentration of COVID-19 cases in a few districts of India and infection ratio. Authors may explain what are the implications of manifold rise of recovery rates until July 2020 on the infection ratio or the implications of increase of average deceased cases by 734 times during studied period.

3) A large focus is given to explaining unadjusted beta coefficients from the results of regression model. Rather, the adjusted figures may be given more writing space. Independent variables are in percent in the linear regression model. Understanding that, the variables effects are interpreted in terms of levels. The interpretation of Table 2 may be revised.

4) In Table 2, authors may check the significance values and the respective confidence interval for unadjusted coefficients. These may be serious errors.

5) The manuscript may conclude strongly about the four determinants connecting it with the trend analysis, infection ratio and hot and cold spots.

6) The manuscript needs an English language editing and flow may be maintained.

Minor comments:

A) In abstract section, (1) the full form of COVID-19 disease may be written correctly, (2) whether authors have used prevalence rate from the crowdsourced data? (3) authors may specific reasons for using data from 14 March 2020, (4) authors may mention crucial information for policy discourse in the conclusion.

B) In Introduction section, (1) line no. 4-6: authors please check the sentence and growth rate of COVID-19 disease in the first three months from Jan to Mar 2020 perhaps, (2) line no. 8-10: check for definite measures, the reduction in doubling time is not apparent from Fig S1, and the second part of the same sentence is contradictory to the first part, (3) line no. 18-19: adults being less susceptible to disease compared to older is based on some assumptions, (4) line no. 22-23: Co-morbid situations are important but reference is missing, (5) line no. 26-27: under-five mortality and nutritional status children may not be associated with the spread of disease.

C) In Outcome variable section, (1) line no. 72, authors check formula.

D) In Spatial Analysis section, (1) line no. 111: description of “low-low” is not given in text.

E) In Result section, (1) line no. 114: authors may check the source used for data if they wish to write “every day since”, (2) line no. 116: Fig S2 and Table S1 provide the same results, (3) line no. 118-121: based on average bi-weekly and base numbers, authors may choose a different growth rate, also Table A1 is not found in text, (4) line no. 129-130: per one hundred thousand versus 100 thousand, authors may use same wording, (5) line no. 146-147: how one confirmed case and positive case are different? (6) line no. 193-194: the interpretation of likelihood in case of linear regression results, (7) line no. 195: which levels of infrastructure has higher effects is not clear from text and Table 2.

F) In Summary and Discussion section, (1) line no. 213: if demographic transition is an appropriate reason, then assumption need to be mentioned in this manuscript, (2) line no. 249: undefined term “unavoidable economic activities”, (3) line no. 259: study design may not be appropriate reason for not explaining co-morbidities in association with disease.

G) In Conclusion section, (1) line no. 267: understanding of disease only by Spatial distribution and its determinants is not supported by the analysis.

6. PLOS authors have the option to publish the peer review history of their article (what does this mean?). If published, this will include your full peer review and any attached files.

Reviewer #1: No

Reviewer #2: No

---

## [Author Response · Author response to Decision Letter 0]

4 Apr 2021

Dear Dr. Goli,

At the outset, we thank you for getting our paper reviewed. We have revised the manuscript according to the suggestions of the reviewers. In the revised version, we have extended the whole analysis to October 31, 2020, interpret the findings and modified the introduction and discussion section accordingly.

We attach a “Point by point reply file” where we answer (and address) all the points raised by the reviewers. Kindly find the attached revised documents.

We sincerely look forward for acceptance of the study in the journal Plosone.

With best regards

All authors

---

## [Decision Letter · Decision Letter 1]

1 Jun 2021

PONE-D-20-34338R1

District level correlates of COVID-19 pandemic in India during March-October 2020

PLOS ONE

Dear Dr. Saikia,

Thank you for submitting your manuscript to PLOS ONE. After careful consideration, we feel that it has merit but does not fully meet PLOS ONE’s publication criteria as it currently stands. Therefore, we invite you to submit a revised version of the manuscript that addresses the points raised during the review process.

ACADEMIC EDITOR: Considering concerns raised by the reviewers, I am sending this paper back to you for necessary amendments. I request the authors to carefully consider both the reviewers suggestions and make desired changes before submitting the revision. 

We look forward to receiving your revised manuscript.

Kind regards,

Srinivas Goli, Ph.D.

Academic Editor

PLOS ONE

Additional Editor Comments (if provided):

Considering concerns raised by the reviewers, I am sending this paper back to authors for necessary amendments. I request the authors to carefully consider both the reviewers suggestions and make desired changes before submitting the revision.

Reviewers' comments:

Reviewer's Responses to Questions

**Comments to the Author**

1. If the authors have adequately addressed your comments raised in a previous round of review and you feel that this manuscript is now acceptable for publication, you may indicate that here to bypass the “Comments to the Author” section, enter your conflict of interest statement in the “Confidential to Editor” section, and submit your "Accept" recommendation.

Reviewer #2: (No Response)

Reviewer #3: (No Response)

2. Is the manuscript technically sound, and do the data support the conclusions?

Reviewer #2: No

Reviewer #3: Partly

3. Has the statistical analysis been performed appropriately and rigorously? 

Reviewer #2: N/A

Reviewer #3: Yes

4. Have the authors made all data underlying the findings in their manuscript fully available?

Reviewer #2: Yes

Reviewer #3: Yes

5. Is the manuscript presented in an intelligible fashion and written in standard English?

Reviewer #2: No

Reviewer #3: Yes

6. Review Comments to the Author

Reviewer #2: The study on “District level correlates of COVID-19 pandemic in India” about exploring the hot and cold spots in terms of districts and finding the socio-economic, demographic and health-related determinants of COVID-19 disease in India has remained more or less at the same position.

Authors may consider these suggestions for future works.

1) The result section is crowded with minor small descriptions such as data description of recovery rates, death rates, which is nowhere connected with any succeeding results, state level and then district level infection ratios, and then again descriptive statistics as shown in Table 1 from which standard deviation for many considered variables are described.

2) The regression results are very poorly summarised. Researchers have ignored the importance of adjusted over unadjusted models. The results clearly warn through the change in P values and coefficients while showing the case of unadjusted and adjusted models. It is percent population in 15-59 years, population density, % urban population, testing, and stunting which are outcomes from regression analysis but not adequately explained in result as well as discussion section of the manuscript.

3) The introduction should be updated with more evidences from the works of literature. The transmission rate of virus of COVID-19 disease is higher during initial times whereas authors have considered it slow based on some works of literature. The widespread of media coverage are not supported by good references. These are controversial and dithering statements given the pandemic.

4) Authors have not taken the numbers and its interpretation seriously. There is ignorance on the part of authors.

5) The term bi-weekly is cumulative or point COVID-19 cases, it is not clear in the text. The bi-weekly numbers of 11 lakhs are shown in Appendix S1 which reveals the bi-weekly peak, as terminology used by authors, was 1244430 in the next week in contrast to the previous week mentioned in text. The Figure 1 and Table S1 shows average numbers; this bi-weekly peak/number is not a meaningful description when original COVID19 data from the API itself provides a more robust measurement.

6) The percent of women whose husbands are away for the last six months is taken as the proxy of out-migration but not taken as one of the variables in analysis. However, percent of marginal workers is considered with the same max and min values of that variable as explained in text.

7) Again, ‘positive (absolute values)’ wording is used in the panel A in Fig 2 and 3. Legends of the maps shown in Fig 2 are inconsistent with Fig 3.

8) The write-up needs to be much more scientific and richer in content which is lacking throughout.

Reviewer #3: The study is important, interesting and generally well written. I have some small recommendations which should be addressedbefore it can be published.

Comments:

1. This study clearly states that it's analysis focuses on the months between July and October 2020. Clearly the covid-19 pandemic is ever-changing and India is clearly in the midst of a catastrophic second wave. The authors have clearly put much effort into their paper so to expect them to include data beyond October 2020 is unreasonable. Instead, the authors should provide a comparison between the first and second waves in their discussion - perhaps comparing case numbers in districts during the first wave to the second? Perhaps discussing which of the factors discussed in this paper may have contirbuted to this devastating second wave?

2. As with a fellow reviewer I'm confused by the use of health related variables focussing only on women (% women with diabetes, % women with cancer, % women consuming tobacco etc). As men are at greater risk of covid-19 why have they been excluded from these analyses? Why also are socioeconomic variables such as religion (% hindu) used? There may be good reasons for this, however the manuscript would benefit from the explanation of such reasons.

3. The manuscript makes no mention of the causative agent of covid-19, SARS-CoV-2. At least one sentence explaining that SARS-CoV-2 is the virus whilst covid-19 is the disease cuased by the virus. Furthermore, the authors must make an effort to discriminate SARS-CoV-2 and covid-19 when discussing spread. For instance, the authors says "COVID-19 has been spreading rapidly in the urban areas". This is inaccurate, SARS-CoV-2 has been spreading.

7. PLOS authors have the option to publish the peer review history of their article (what does this mean?). If published, this will include your full peer review and any attached files.

Reviewer #2: No

Reviewer #3: No

---

## [Author Response · Author response to Decision Letter 1]

31 Aug 2021

The study on “District level correlates of COVID-19 pandemic in India” about exploring the hot and cold spots in terms of districts and finding the socio-economic, demographic and health-related determinants of COVID-19 disease in India has remained more or less at the same position. 

Reviewer comments:

Authors may consider these suggestions for future works.

The result section is crowded with minor small descriptions such as data description of recovery rates, death rates, which is nowhere connected with any succeeding results, state level and then district level infection ratios, and then again descriptive statistics as shown in Table 1 from which basic statistics for many considered variables are described.

Response – Thank you. We agree with the reviewer that we have done sufficient analysis to explore district level COVID-19 pattern. Therefore, we have modified our objective (Line 116-118). We kept the national level analysis (Figure 1) as background information. The state level analysis is presented only in the appendix with a short description of these results in line number 216-228. 

The regression results are very poorly summarised. Researchers have ignored the importance of adjusted over unadjusted models. The results clearly warn through the change in P values and coefficients while showing the case of unadjusted and adjusted models. It is percent population in 15-59 years, population density, % urban population, testing, and stunting which are outcomes from regression analysis but not adequately explained in result as well as discussion section of the manuscript. 

Response – We have updated regression results (the unadjusted model in the line number 308-325 and then the adjusted model from 326 to 341 of the manuscript. All the significant variables were explained adequately in the revised manuscript.

The introduction should be updated with more evidences from the works of literature. The transmission rate of virus of COVID-19 disease is higher during initial times whereas authors have considered it slow based on some works of literature. The widespread of media coverage are not supported by good references. These are controversial and dithering statements given the pandemic. 

Response – We have updated the introduction section. We have not refered any media article; we rather refered articles by researchers. Due to nation-wide lockdown, the transmission rate was slow at the beginning of COVID19 in India. We have also included some good references. 

Authors have not taken the numbers and its interpretation seriously. There is ignorance on the part of authors.

Response – We have revised the write-up in the revised draft. 

The term bi-weekly is cumulative or point COVID-19 cases, it is not clear in the text. The bi-weekly numbers of 11 lakhs are shown in Appendix S1 which reveals the bi-weekly peak, as terminology used by authors, was 1244430 in the next week in contrast to the previous week mentioned in text. The Figure 1 and Table S1 shows average numbers; this bi-weekly peak/number is not a meaningful description when original COVID19 data from the API itself provides a more robust measurement. 

Response – The term bi-weekly is non-cumulative. The number is correct and text was modified accordingly. The numbers have been revised in the text (line number 83-85 of the manuscript). The term bi-weekly in Figure 1 and Table S1 was not cumulative cases. It showed the average number of cases per 14 days. We showed 14 days average cases as when we present per each day, it becomes relatively cumbersome.

The percent of women whose husbands are away for the last six months is taken as the proxy of out-migration but not taken as one of the variables in analysis. However, percent of marginal workers is considered with the same max and min values of that variable as explained in text. 

Response – We have removed the proxy of out-migration variable (the percent of women whose husbands are away for the last six months) from tables and text as it has not been included in regression analysis.

Again, ‘positive (absolute values)’ wording is used in the panel A in Fig 2 and 3. Legends of the maps shown in Fig 2 are inconsistent with Fig 3.

Response – We have revised this legend by “COVID19 Confirmed” and used the “Absolute number of COVID-19 cases” in panel A of both Figure 2 & 3.

The write-up needs to be much more scientific and richer in content which is lacking throughout. 

Response: We have modified the write-up.

Reviewer #3: The study is important, interesting and generally well written. I have some small recommendations which should be addressed before it can be published.

Comments

1. This study clearly states that its analysis focuses on the months between July and October 2020. Clearly the covid-19 pandemic is ever-changing and India is clearly in the midst of a catastrophic second wave. The authors have clearly put much effort into their paper so to expect them to include data beyond October 2020 is unreasonable. Instead, the authors should provide a comparison between the first and second waves in their discussion - perhaps comparing case numbers in districts during the first wave to the second? Perhaps discussing which of the factors discussed in this paper may have contributed to this devastating second wave?

Response If we discuss first and second wave of COVID-19 together, we need to discuss it based on analysis, speculations will not be a good strategy. However, adding new analysis for the second wave in this study is beyond the scope of the paper. Covid-19 situation has been ever changing, and hence we want to restrict our analysis within the first wave.

Yet, as per reviewer’s suggestion, we compared the situation in the second wave as a part of discussion (Line number 402-424 of the manuscript). 

2. As with a fellow reviewer I'm confused by the use of health-related variables focussing only on women (% women with diabetes, % women with cancer, % women consuming tobacco etc). As men are at greater risk of covid-19 why have they been excluded from these analyses? Why also are socioeconomic variables such as religion (% Hindu) used? There may be good reasons for this, however the manuscript would benefit from the explanation of such reasons.

Response – The design of the regression analysis presented here is not an individual level of analysis, rather a group level (district level analysis. Our independent variables are taken at macro level (i.e., district level) to determine overall health status and health care service utilization status. Therefore, fasting glucose or use of tobacco among women is representation to the overall health status of total population of the district. Similarly, immunization and childhood illness are taken to represent overall health status and health care service utilization. Percent of Hindu population is used as an indicator of majority population (which indirectly reflect the presence of minority population) at district level. 

3. The manuscript makes no mention of the causative agent of covid-19, SARS-CoV-2. At least one sentence explaining that SARS-CoV-2 is the virus whilst covid-19 is the disease caused by the virus. Furthermore, the authors must make an effort to discriminate SARS-CoV-2 and covid-19 when discussing spread. For instance, the authors say "COVID-19 has been spreading rapidly in the urban areas". This is inaccurate, SARS-CoV-2 has been spreading.

Response – We have mentioned SARS-CoV-2 in the introductory part of the revised manuscript (Line no. 70-73

---

## [Editor Report · Decision Letter 2]

7 Sep 2021

District level correlates of COVID-19 pandemic in India during March-October 2020

PONE-D-20-34338R2

Dear Dr. Saikia,

We’re pleased to inform you that your manuscript has been judged scientifically suitable for publication and will be formally accepted for publication once it meets all outstanding technical requirements.

Kind regards,

Srinivas Goli, Ph.D.

Academic Editor

PLOS ONE

Additional Editor Comments (optional):

Authors revised the paper according to reviewers comments, thus I am recommending this paper.
---

## [Editor Report · Acceptance letter]

23 Sep 2021

PONE-D-20-34338R2 

District level correlates of COVID-19 pandemic in India during March-October 2020 

Dear Dr. Saikia:

I'm pleased to inform you that your manuscript has been deemed suitable for publication in PLOS ONE. Congratulations! Your manuscript is now with our production department. 

Kind regards, 

on behalf of

Dr. Srinivas Goli 

Academic Editor

PLOS ONE